# Physical Activity, Sleep, and Demographic Patterns in Alaska Native Children and Youth Living in Anaktuvuk Pass

Vernon Grant [1], Deborah Mekiana [2] and Jacques Philip [3,*]

1. Center for American Indian and Rural Health Equity, Montana State University, Bozeman, MT 59717, USA
2. Department of Alaska Native Studies and Rural Development, University of Alaska, Fairbanks, AK 99775, USA
3. Center for Alaska Native Health Research, University of Alaska Fairbanks, Fairbanks, AK 99775, USA
* Correspondence: jphilip@alaska.edu

**Abstract:** Physical activity (PA), sleep, and weight are important factors for youth health. However, data about these factors are unknown in youth living in isolated Alaska Native communities. This study aims to assess PA, sleep, height and weight in elementary through high school students living in Anaktuvuk Pass. Fourteen children (<12) and 24 youths (12–20) volunteered to participate in this study. PA and sleep data were collected with actigraphy. Height and weight were assessed with standard procedures. Demographics were collected via survey. Results show that 10.53% and 18.42% of participants were overweight and obese, respectively. Average bedtime was 00:15 am and wake time 08:23 am. Total sleep time was 498.21 min. Participants averaged 477.64 min in sedentary activity, 297.29 min in light activity, 150.66 min in moderate activity, and 18.05 min in vigorous activity. Adjusted models suggest that high school students engage in significantly more sedentary activity, and significantly less light, moderate, and vigorous activity compared to those in middle and elementary school. All students engaged in less moderate and vigorous activity on the weekend compared to the weekday. Data suggest that as children age they become more sedentary. Future studies should focus on increasing daily PA in high school students while considering other obesogenic factors.

**Keywords:** youth; physical activity; sleep; Alaska Native community





## 1. Introduction

Alaska Native people in Anaktuvuk Pass have historically had a very active and healthy lifestyle that is related their subsistence lifestyle. However, after colonization they began experiencing important cardio-metabolic health disparities [1]. Changes in culture and lifestyle were particularly recent and rapid for Alaska Native people, with deleterious consequences [2,3]. One of the authors (DM) shares some of this history in the community where the study took place.

> *"Anaktuvuk Pass was settled in 1949. It is said there were 3 reasons why the seven families chose this particular valley. The Brooks Range mountains provide numerous streams of melting glacial water. The willows were once large and plentiful for firewood. The caribou follow their ancestral trails in the valley each spring and fall during their migration to or from the winds of the North Slope.*

> *We are located in the north central area of the Brooks Range. The closest town with a hospital, store, and restaurant is 250 air miles south. To the north, the closest town is approximately 250 air miles across the Beaufort Sea. Our home is in the mountains north of the arctic circle 2000 feet above sea level. Our population is approximately 300 people, with the majority Inupiaq. More specifically, Nunamiut or the people of the land.*

> *The Nunamiut have been said to be the last nomadic people of North America. My father was part of these last nomadic Indigenous people. He passed away when I was a child*

*but his hunting partner/best friend has shared some stories of their travels with me. They would walk miles per day looking for caribou. Caribou has always been our main sustenance. Snaring rabbits, squirrels, or ptarmigan on their journeys would provide much needed food for these long walks searching for caribou. Looking for food in the Brooks Range mountains can be difficult. They had to traverse the rough terrain of tundra, foothills, and mountain peaks that can reach thousands of feet high. The tundra consists of tussocks with potholes that can be several feet deep, holding water that does not move, infested with mosquito larvae. In the summer, walking was avoided in the lower altitude areas as the mosquito infestation would be terrible. The tundra on the foothills of the mountains are full of rocks and boulders. Maneuvering through this landscape can be similar to maneuvering an obstacle course, always cautious not to get your foot or leg stuck in a crevice while jumping or hopping from boulder to boulder, which can severely damage your body. The less challenging tundra is flat with small rocks scattered around the dwarfed shrubbery created by lack of growth due to the Arctic climate. These more merciless areas are located on flat spots on the top of a foothill located at the bottom of the mountains. The amount of physical work that is needed for living a subsistence lifestyle, the Nunamiut hunters have been and continue to be in very good physical condition. My father and his paanaq (partner)—specifically hunting partner in this context—roamed the Brooks Range always seeking food. Dogs were utilized as they traveled long distances through this rough terrain. My father and his paanaq harnessed dog packs onto their dogs which carried supplies and the dismembered bodies of caribou if they were lucky to harvest any. The dog packs were constructed out of caribou hides, hand-tanned, and sewn together with the split, dried sinew of the caribou. One caribou skin dog pack is preserved at the Simon Paneak Memorial Museum in Anaktuvuk Pass and was made by my father.*

*My cousins and I were told as children that dogs would always help us if we took good care of them. There are stories of Simon Paneak (whom the museum is named after) who sat and picked the dried feces from his dogs behind because he wanted his dogs be clean. Up until the mid-1980s when the Gates of the Arctic National Park and Preserve bordered three sides of Anaktuvuk Pass, the Nunamiut used dogs for pulling sleds in the winter and pack in the summer. When the park was created, the Nunamiut were no longer allowed to feed their dogs solely caribou meat. There is no road into Anaktuvuk Pass so all store bought supplies have to be brought in by bush plane. A 50 pound (22.7 kg) bag of dog food may cost \$60 but when you add on the \$1 per pound freight charge, your 50 pound bag of dog food now costs \$110. I have two dogs in Anaktuvuk Pass that consume about 150 pounds (68 kg) of dog food a month. This is \$330 worth of dog food a month for two dogs."*

Explorer Helge Ingstad, who spent a winter with the Nunamiut tribe in 1949, noted that children were very healthy and happy, and hunters had remarkable physical abilities [4]. Many artifacts of the Nunamiut ancestral lifestyle are preserved in the Community of Anaktuvuk Pass Museum. Contemporary residents of Anaktuvuk Pass likely do not have the same level of physical activity (PA) as their predecessors, but until now no data were available to quantify this. This study is part of a broader effort to promote PA among youth in Alaska Native communities by developing the practice of cross-country skiing and incorporating the use of sled dogs (skijoring). This strategy may have additional mental health benefits, while minimizing the financial burden compared to sledding activities that would require more animals. The picture in Figure 1 illustrates the environment in which this study took place. It is important to note that sunlight in Anaktuvuk Pass varies between two hours at the heart of winter to 24 h a day at the peak of the summer. During data collection in April, sunrise occurred at 5:50 am and sunset at 10:20 pm. Because of low sunlight exposure and a nutritional transition, Alaska Native children exhibit high rates of vitamin D deficiency [5].

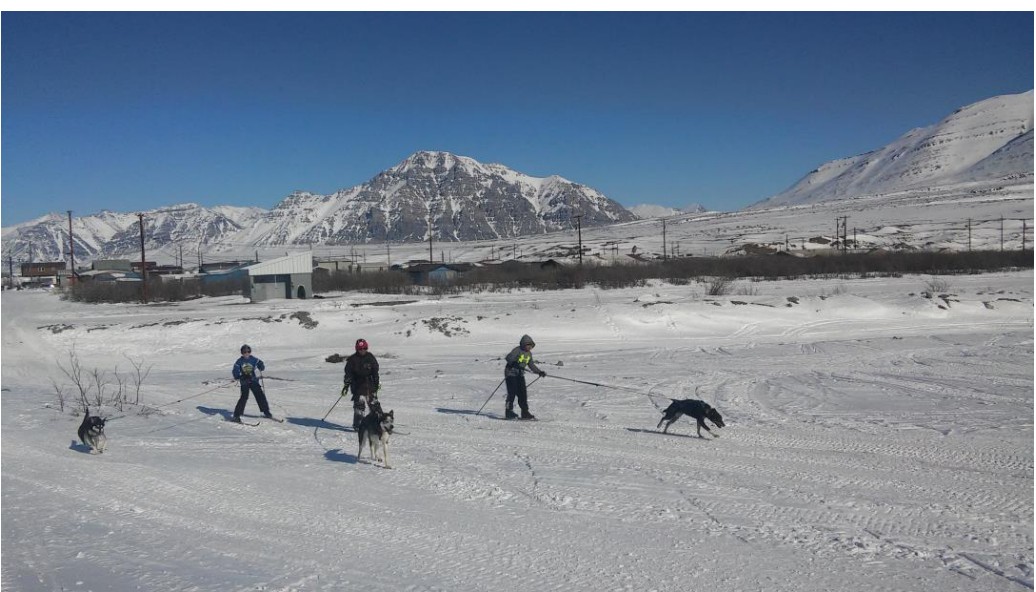

**Figure 1.** Youth from Anaktuvuk Pass participating in the skijoring program. In this arctic environment, April temperatures can be as low as −20 °F (−30 °C) and the closest town is 250 miles (400 km) south with no road or navigable waterway. Copyright Jacques Philip, 2017.

PA is a crucial modifiable behavior to enhance health. This is important to understand in American Indian and Alaska Native (AIAN) communities due to the alarmingly high rates of obesity [6–8] and the subsequent high risk of chronic disease [9–11]. In contrast, individuals engaging in adequate daily PA have profound health benefits that have been well established [12,13]. Unfortunately, most individuals do not obtain the recommended 60 min of PA per day [14]. A systematic review of the literature to determine a report card on how well the United States is doing with PA assigned a failing grade for overall PA levels in children and youth attaining 60 min or more of moderate-to-vigorous physical activity (MVPA) [15]. Other studies report that children spend 70% of their time in sedentary activity [16]. Accordingly, little work has been undertaken to objectively measure PA in AIAN children. A systematic review of PA in American Indian (AI) youth reports only four studies that objectively measure PA [17]. One study does not report outcome data [18], and two studies report outcome data in vector magnitude per hour, which is not comparable [19,20]. Other works undertaken with AI youth assessing objective measures do not report cross-sectional PA patterns [21], and one study with urban AI youth reports an average of 178.80 min of MVPA per day [22]. Moreover, pedometer and accelerometer data in Yup'ik adults report an average of 7531 steps per day [23] and an average of 201.90 min of MVPA/day [24], respectively. The sparse amount of work supports the need to better understand PA patterns in AIAN youth, especially Nunamiut in Anaktuvuk Pass.

PA patterns have been shown to dramatically shift from the weekday to the weekend. In the general population, studies report that youth engage in less PA on the weekend compared to the weekday, with decreases ranging from 6.07 min to as high as 18.00 min [16,25–27]. Sedentary activity has also been reported to significantly shift from weekday to weekend [26]. Studies show that children obtain the most MVPA during weekday school time [28]. However, little is known about PA patterns in AIAN youth. Brusseau et al. [29] collected PA with pedometry in 5th–6th grade children in a southwestern AI community and found average weekday and weekend steps at 11,891 and 7370, with an average difference of 4521 steps. Grant et al. [22] conducted a cross-sectional study in urban AI youth with accelerometry and found a significant decrease in light, moderate, vigorous and MVPA, and a significant increase in sedentary activity on the weekend compared to the weekday. These studies demonstrate the variability in PA from the weekday to weekend and the sparse understanding about PA patterns in AIAN youth.

Arguably, sleep research is still in its infancy and sleep patterns in youth are vaguely understood. Understanding sleep patterns is important as children aged 6 years to 17 years report high levels of inadequate sleep [30]. This is a particular concern for AIAN youth, as previous work suggests that minority youth sleep less than Caucasian youth [31]. Studies in youth report sleep duration ranges between 7.30 h and 9.40 h, bedtime ranges between 10:45 pm and 11:18 pm, and waketime ranges between 6:45 am and 7:51 am, respectively [22,32–35]. Understanding youth sleep patterns is critical in order to maintain regular waketime and nightly sleep duration.

Sleep changes mirror PA in terms of shifts from weekday to weekend. Several studies report increased sleep duration on the weekend compared to the weekday, with shifts ranging from 12.00 min to as much as an hour [26,36–40]. Another study assessing children aged 12–18 discovered weekend sleep shifts of 111.00 min, just shy of two hours [41]. Moreover, a study with youth of mean age 14 years reported that 66% of the participants had a weekday to weekend shift of sleep > 2 h in favor of the weekend [35]. According to the literature, there is one study that reports weekday to weekend shifts in sleep with AIAN youth in this age group. Grant et al. [22] conducted a study with urban AI children in the 6th–8th grade and found a weekend increase of 42.50 min. These data in youth underscore the variability in sleep and demonstrate that weekday to weekend sleep is not maintained.

The Center for Alaska Native Health Research (CANHR) study has carried out work with Alaska Native populations focused on obesity genetics, community nutrition and PA, and cultural–behavioral health measurement and development [42]. However, much of this work reports on risk factors and the prevalence of metabolic syndrome in Southwest Alaskan tribes [43,44]. This study builds upon the existing work that has been carried out, with a specific focus on sleep and PA in Nunamiut youth. Therefore, the purpose of this study is to describe PA, sleep, height and weight variables in elementary through high school youth and children living in Anaktuvuk Pass in Alaska.

## 2. Materials and Methods

### 2.1. Study Design and Participants

This work was a collaboration between CANHR at the University of Alaska Fairbanks (UAF), the community of Anaktuvuk Pass, the Nunamiut School, and the Alaska Nordic ski program. It was guided by community-based participatory research principles; in particular, the project leader obtained permission from the tribal council to submit the proposal and established a community steering committee charged with guiding the research and insuring its cultural relevance. The long-term goal of the project was to establish dogjoring activities (a dog pulling a skier or hiker) as a supplement to an existing cross-country ski program provided by an outside entity to increase youth motivation to engage in PA. The assumption is that Nunamiut children are naturally attracted to interact with dogs, with the added bonus that Anaktuvuk Pass has a rich history of using dogs for packing and sledding, thus increasing the cultural relevance of the program. The study presented here is a cross-sectional pilot assessing PA and sleep in Nunamiut children. All procedures were approved by the UAF Institutional Review Board (IRB), under protocol # 1089068.

### 2.2. Instruments and Variables

Data collection occurred during one week in April, as the Alaska Nordic ski program, including dogjoring activities, was ongoing in the community. The research team, composed of the project leader and three ski/skijoring coaches, traveled to Anaktuvuk Pass for a week. All elementary to high school students aged 12–18 were invited, and 38 participated. The ski program was integrated in physical education classes during winter and spring. In addition, during the week of data collection, the dogjoring activities supplemented the skiing activities, allowing each student to participate in one hour of daily skiing/skijoring during the school day and with the option of one to two additional hours after school. Other after school activities, such as basketball, were also available to students.

Overall, the skijoring activities did not represent an unusual amount of PA for the youth as they typically participate in various afterschool activities. Each youth participant signed an assent form and provided parental consent, which was facilitated by a high school student (i.e., research assistant) who received IRB training. Students were called in small groups to a station in the school, and were asked to complete a short demographic questionnaire, participate in anthropometric measurements, and an accelerometer was administered. Each student received a USD 25 gift card for Amazon upon enrollment, and another USD 25 gift card for Amazon when they returned the accelerometer a week later.

### 2.2.1. Physical Activity and Sleep

PA and sleep were assessed by the ActiGraph GT3X+ accelerometer (Actigraph Pensacola, FL, USA), a valid measure of activity energy expenditure in youth [45]. Accelerometers were fitted on the non-dominant wrist. The devices' frequency was set at 90 Hz, which allows for maximum data density. Participants were instructed to wear the devices 24 h/day for seven days starting on Monday. Weekend PA periods were defined as occurring Saturday or Sunday and weekend sleep periods include Friday and Saturday night. For PA, a participant must have three valid weekdays and one valid weekend day, with a day having a minimum of 480.00 min (eight hours) of wear time. For sleep, a participant must have three weekday and one weekend sleep period.

The PA cutpoints were based on counts per five second epochs on the Y axis: sedentary 0.00 to 35.00, light 36.00 to 360.00, moderate 361.00 to 1129.00, and vigorous 1130.00 and above [46]. Sleep measures are: in bedtime (IBT) and out of bedtime (OBT), as identified by automatic sleep detection in Actilife, time in bed (TIB), defined as the difference between OBT and IBT, and sleep efficiency as the number of sleep minutes divided by TIB [47,48].

### 2.2.2. Body Composition

Height was measured to the nearest 0.10cm with a portable stadiometer (Perspective Enterprises, PE-AIM-101; Portage, MI, USA), weight was measured to the nearest 0.10 kg with a portable scale (Seca Model 876; Chino, CA, USA), and waist circumference was measured to the nearest 0.10 cm with a plastic tape (Seca Model 201; Chino, CA, USA). All measurements were recorded three times for each participant. Measures were repeated if they were not within 0.20 units (e.g., 0.20 kg for weight). BMI percentiles and z-scores (BMI-Z) were calculated in SAS using the Centers for Disease Control and Prevention calculator [49]. The analysis was unable to compute BMIZ and BMI percentiles for one participant because he was over 20 years old, which is above the limit for those measures.

### 2.2.3. Demographics

Participants completed an eight-item demographic survey that assessed age, grade, gender, race and ethnicity (i.e., Alaska Native). Traditional and Western lifestyle were assessed with three options: "Not at all", "Some" and "A lot". These variables measured how much an individual identified with living a traditional Nunamiut lifestyle or identified with living a Western colonial lifestyle.

### 2.3. Procedure

ActiLife V6.13.40 software was used to download accelerometer raw data and generate activity counts on three axis, with 60.00 s epochs for sleep and five second epochs for PA. Sleep and PA data are traditionally analyzed in 60.00 s epochs; however, recent work showed that five second epochs are better at detecting short bursts of vigorous activity in youth [46]. A customized version of the Choi algorithm [50] was used to detect non-wear periods with the following parameters: minimum length: 60.00 min (the minimum length of time for consecutive zero counts to be considered a non-wear time interval), small window length: 30.00 min (the minimum length of up/down-stream time window for consecutive zero counts required before and after the artifactual movement interval to be considered a non-wear time interval), and spike tolerance: two minutes (the maximum

length of time interval allowed for the artifactual non-zero counts during a non-wear time interval). Ignore wear periods were defined as less than 30.00 min.

The Sadeh algorithm on one minute epoch data was used to score sleep [48,51] as well as the automatic sleep period detection feature in Actilife, using a custom Tudor-Locke [52] algorithm with the following parameters: minimum sleep period of 160.00 min, waketime definition of 30.00 min of waketime, bedtime definition of five minutes sleep time, max time between bedtime and waketime of 1440.00 min, and 15.00 min of non-zero epochs. Due to the algorithm requiring several minutes of 'sleep' epochs at the start of a sleep period, the latency will be zero for any sleep periods that are found. Manual adjustments after automatic sleep detection were made to remove nap periods and join sleep periods that occurred the same night. After all the pre-processed accelerometry datasets were computed, the data were then imported into R and merged with the demographic and measurement data for analysis.

### 2.4. Data Analysis

Summary statistics and inference were computed in R v. 4.0. Summary statistics of the time of day were computed using the circular package v. 0.4-93. ANOVA tests were used to compare numerical differences, e.g., MVPA and a time specific ANOVA from the circular package (aov.circular function) was used to test time differences. Linear mixed models were constructed to test differences in PA with covariates and to take advantage of the within and across participant variability (each participant had multiple days). Diagnostics plots of linear mixed models showed that the only assumption not met was a deviation from normality. Therefore, Linear Quantile Mixed Models (LQMMs) were fit instead, using the lqmm R package v. 1.5.2.

## 3. Results

### 3.1. Demographics and Body Composition

A total of 38 children participated in this study (n = 27 males (71.05%); Table 1). Grade breakdown consisted of 17 (44.74%) in elementary school, nine (23.68%) in middle school, and 12 (31.58%) in high school, with a mean age of 12.82 ± 2.80. Body composition found that four (10.53%) were overweight and seven (18.42%) obese.

**Table 1.** Demographic and anthropometric statistics (n = 38).

| Variables | N (%) or Mean ± sd |
|---|---|
| Age | |
| [9–11] | 14 (36.84) |
| [12–15] | 17 (44.74) |
| [16–20] | 7 (18.42) |
| Gender | |
| Male | 27 (71.05) |
| Female | 11 (28.95) |
| Grade | |
| Elementary [4–6] | 17 (44.74) |
| Middle school [7,8] | 9 (23.68) |
| High school [9–12] | 12 (31.58) |
| Race/Ethnicity | |
| Inupiaq | 34 (89.47) |
| White | 4 (10.53) |
| Traditional lifestyle | |
| Not at all | 1 (2.63) |
| Some | 24 (63.16) |
| A lot | 13 (34.21) |

**Table 1.** *Cont.*

| Variables | N (%) or Mean ± sd |
|---|---|
| Western lifestyle | |
| Not at all | 3 (7.89) |
| Some | 21 (55.26) |
| A lot | 14 (36.84) |
| Combined lifestyle score [2–6] | |
| 3 | 1 (2.63) |
| 4 | 16 (42.11) |
| 5 | 18 (47.4) |
| 6 | 3 (7.89) |
| BMI percentile [1] | |
| Normal weight | 26 (68.42) |
| Overweight | 4 (10.53) |
| Obese | 7 (18.42) |
| Missing | 1 (2.63) |
| BMIZ | |
| Elementary [4–6] | 0.64 ± 1.01 |
| Middle school [7,8] | 0.58 ± 0.68 |
| High school [9–12] | 1.02 ± 1.16 |
| All | 0.74 ± 0.98 |

[1] BMI percentiles: 0–5th = underweight; 5th–85th = normal weight; 85th–95th = overweight; ≥95th = obese.

### 3.2. Sleep Patterns

Participants' average bedtime was 00:15 am and waketime 08:23 am, with an average total sleep time of 498.21 min (Table 2). Shifts were observed from weekday to weekend. For instance, weekday bedtime was 00:04 am and weekend bedtime 00:43 am (Δ 39 min; *p*-value = 0.061), and weekday waketime was 08:00 am and weekend waketime 09:25 am (Δ 85 min; *p*-value ≤ 0.001). Weekday total sleep time was 480.46 min and weekend total sleep time was 541.44 min (Δ 60.98 min; *p*-value ≤ 0.01).

**Table 2.** Descriptive statistics of sleep (n = 33).

| Variables | Weekday (N or Mean ± sd) | Weekend (N or Mean ± sd) | All (N or Mean ± sd) | *p*-Value |
|---|---|---|---|---|
| Number of sleep periods (n) | 151 | 62 | 213 | |
| Average in bed time (hh:mm) | 00:04 AM ± 00:35 | 00:43 AM ± 00:38 | 00:15 AM ± 00:36 | 0.061 |
| Average out of bed time (hh:mm) | 08:00 AM ± 00:23 | 09:25 AM ± 00:34 | 08:23 AM ± 00:28 | ≤0.001 |
| Average time in bed (min) | 480.46 ± 139.23 | 541.44 ± 183.09 | 498.21 ± 155.35 | 0.009 |
| Average sleep efficiency (%) | 86.24 ± 7.12 | 85.34 ± 8.13 | 85.98 ± 7.42 | 0.424 |
| Average wake after sleep onset (min) | 66.57 ± 42.96 | 80.50 ± 55.73 | 70.62 ± 47.33 | 0.051 |
| Average period sleep total time (min) | 480.19 ± 139.28 | 540.69 ± 183.19 | 497.80 ± 155.4 | 0.010 |
| Average number of awakenings (n) | 20.63 ± 8.29 | 23.79 ± 12.14 | 21.55 ± 9.65 | 0.030 |
| Average length of awakenings (min) | 3.18 ± 1.4 | 3.40 ± 1.78 | 3.25 ± 1.50 | 0.345 |
| Average period activity count (n) | 30,805.54 ± 24,175.20 | 35,319.94 ± 27,155.57 | 32,119.59 ± 25,098.4 | 0.234 |
| Average movement index (MI) [1] | 11.89 ± 4.24 | 12.79 ± 5.56 | 12.15 ± 4.67 | 0.203 |
| Average fragmentation index (FI) [2] | 10.46 ± 8.06 | 10.60 ± 7.79 | 10.50 ± 7.96 | 0.904 |
| Average sleep fragmentation index (SFI) [3] | 22.34 ± 10.4 | 23.4 ± 11.50 | 22.65 ± 10.71 | 0.519 |

[1] MI = percentage of epochs with y-axis counts greater than zero in the sleep period. [2] FI = percentage of one minute periods of sleep vs. all periods of sleep during the sleep period. [3] SFI = sum of MI and FI.

### 3.3. Physical Activity Patterns

Participants on average engaged in 477.64 min sedentary activity, 297.29 min light activity, 150.66 min moderate activity, 18.05 min vigorous activity, and 168.71 min MVPA (Table 3). Shifts were observed from weekday to weekend. For instance, sedentary activity was 474.43 min during weekday and 485.66 min during weekend (Δ 11.23 min;

*p*-value = 0.58), light activity was 300.67 min during weekday and 288.84 min during weekend (Δ 11.83 min; *p*-value = 0.298), moderate activity was 157.46 min during weekday and 133.66 min during weekend (Δ 23.80 min; *p*-value = 0.08), vigorous activity was 21.30 min during weekday and 9.92 min during weekend (Δ 11.38 min; *p*-value ≤ 0.001), and MVPA was 178.77 min during weekday and 143.58 min during weekend (Δ 35.19 min; *p*-value ≤ 0.001).

**Table 3.** Descriptive statistics of physical activity (n = 31).

| Variables | Weekday (Mean ± sd or %) | Weekend (Mean ± sd or %) | All (Mean ± sd or %) | *p*-Value |
|---|---|---|---|---|
| Activity cut-points | | | | |
| Average sedentary time/day (min) | 474.43 (129.51) | 485.66 (140.04) | 477.64 (132.36) | 0.580 |
| Average proportion of sedentary time/day (%) | 49.66 | 52.91 | 50.56 | |
| Average light activity time/day (min) | 300.67 (60.00) | 288.84 (101.85) | 297.29 (74.32) | 0.298 |
| Average proportion of light activity time/day (%) | 31.47 | 31.47 | 31.47 | |
| Average moderate activity time/day (min) | 157.46 (52.95) | 133.66 (69.52) | 150.66 (58.99) | 0.008 |
| Average proportion of moderate activity time/day (%) | 16.48 | 14.56 | 15.95 | |
| Average vigorous activity time/day (min) | 21.30 (16.71) | 9.92 (12.95) | 18.05 (16.52) | ≤0.001 |
| Average proportion of vigorous activity time/day (%) | 2.23 | 1.08 | 1.91 | |
| Average MVPA time/day (min) | 178.77 (64.23) | 143.58 (78.00) | 168.71 (70.09) | ≤0.001 |
| Average proportion of MVPA time/day (%) | 18.71 | 15.64 | 17.86 | |

Note: Numbers are adjusted for wear time (missing data).

*3.4. Quantile Linear Regression Models*

Adjusted PA models show that high school youth engaged in significantly more sedentary activity (β = 122.55 min; *p*-value ≤ 0.001), and significantly less light (β = −54.27 min; *p*-value ≤ 0.01), moderate (β = −67.50 min; *p*-value ≤ 0.001), and vigorous PA (β = −10.28 min; *p*-value ≤ 0.001), compared to middle school and elementary school youth and children (Table 4). Results also show that all participants engaged in significantly less moderate (β = −32.15 min; *p*-value ≤ 0.001) and vigorous PA (β = −9.63 min; *p*-value ≤ 0.001) on the weekend compared to the weekday. Boys engaged in slightly more PA and less sedentary time than girls; however, none of the differences were statistically significant.

**Table 4.** Quantile linear mixed regression models of different PA activities adjusted by gender, grade, and weekend.

| | Average Sedentary Time/Day (min) | | Average Light Time/Day (min) | | Average Moderate Time/Day (min) | | Average Vigorous Time/Day (min) | |
|---|---|---|---|---|---|---|---|---|
| | β | *p*-Value | β | *p*-Value | β | *p*-Value | β | *p*-Value |
| Intercept | 425.62 | ≤0.001 | 314.46 | ≤0.001 | 200.64 | ≤0.001 | 21.24 | ≤0.001 |
| Female | 35.27 | 0.39 | −4.40 | 0.82 | −20.04 | 0.14 | −1.90 | 0.52 |
| High School * | 122.55 | ≤0.001 | −54.27 | ≤0.01 | −67.50 | ≤0.001 | −10.28 | ≤0.001 |
| Weekend | 12.70 | 0.43 | −17.60 | 0.27 | −32.15 | ≤0.001 | −9.63 | ≤0.001 |

* Grade 9–12 vs. all other participants

**4. Discussion**

The purpose of this study was to describe PA and sleep patterns in school-aged youth and children living in an isolated Alaska Native community in Anaktuvuk Pass. Results show that high school participants engage in significantly more sedentary activity and significantly less light, moderate, and vigorous activity compared to all other participants. In addition, all participants had shifts in activity patterns from weekday to weekend, such that moderate activity and vigorous activity were significantly less on the weekend. This held true for sleep, which showed shifts in bedtime, waketime, and total sleep time with more minutes during the weekend compared to the weekday.

The PA patterns observed in this study are consistently documented in the literature, and it appears that as children age to youth, they move less. For example, studies show that youth primarily engage in PA during school [53] and engage in more sedentary activity, such as increased screen time, outside of school [54,55]. This trend is also observed in longitudinal studies, with a decrease in PA and increase in screen time as children age [56–59]. This is an important consideration as research has shown a relationship between decreased PA and increase in BMI [60], which worsens as youth age in to adulthood [61]. These trends have important implications for the community of Nunamiut. First, the youth live in an isolated, underserved community with limited PA opportunities. Second, Nunamiut derive from generations of hunters and gatherers that possessed immense physical abilities attributed to their subsistence lifestyle. Colonialism has severely disrupted that way of life and caused deleterious effects on the PA of Nunamiut. Finally, this work is important to highlight the needs that exist in this community, specific to finding strategies to increase PA in this population.

This sample of Nunamiut youth and children engaged in significantly less moderate and vigorous activity on the weekend compared to the weekday. This study aligns with much of the literature assessing PA levels in the general population. Several studies report shifts ranging from 6.70 min to 18.00 min in terms of less MVPA on the weekend compared to weekday [16,25–27]. This trend also holds true for sedentary activity, where others report that sedentary activity increases on the weekend compared to the weekday [26]. These patterns were also observed in urban AI youth, where sedentary activity increased and MVPA decreased on the weekend compared to the weekday [22]. It appears that youth are obtaining the majority of their PA on the weekday during school. For instance, fewer youth achieve $\geq$60 min of MVPA on weekends compared to weekdays (46% vs. 22%; $p \leq 0.001$), which equates to fourfold higher odds of not obtaining $\geq$60 min of MVPA on the weekend [16]. Because of the small sample size, the analysis was unable to assess gender differences, which is an important consideration. Studies report that boys not only engage in more MVPA than girls, but demonstrate less of a shift from the weekday to weekend than girls [28]. This has implications for researchers working to design PA interventions, suggesting tailoring the design to ensure girls are not only increasing activity levels, but also mitigating the PA decline on the weekend. It is important to mitigate weekday to weekend shifts because a decrease in MVPA and increase in sedentary activity show that youth have higher odds of obesity [26].

The overall average MVPA of 168 min/day is largely above the current recommendation of 60 min/day [14], indicating that youth and children from Anaktuvuk Pass may have retained some of the high activity behavior of their elders. A high level of fitness was confirmed by an informal qualitative assessment administered by the ski coaches (data not shown). Directly comparing activity levels between studies is challenging because of the differences in accelerometers, epochs, cut-points, and treatment of missing data. However, a study by Kim et al. [62] used a similar methodology on a sample of US youth and children of comparable ages and reported an average MVPA of $106.90 \pm 1.80$ min/day. In contrast, 40% of the participants in this study are overweight or obese compared to approximately 20% in the general population, highlighting the importance of other obesogenic factors. An important consideration is the introduction of sugary foods and drinks, while Nunamiut had been eating mostly a protein/fat diet for generations, which suggests that they have not adapted to sugary diets. In addition, the stress imposed by colonization has been proposed as contributing to obesity in Alaska Native people [23].

The sample of Nunamiut youth and children in this study demonstrated shifts amongst different sleep variables from the weekday to weekend. Specifically, weekday bedtime was 00:04 am and weekend bedtime was 00:43 am, a bedtime shift of 39.00 min later on the weekend. The same trends were observed for waketime, such that weekday waketime was 8:00 am and weekend waketime was 9:25 am, 1 h 25 min of sleeping in on the weekend. Moreover, total sleep time was 60.50 min more on the weekend compared to weekday. The literature shows similar patterns in shifts from weekday to weekend in total sleep time of

14.40 min to 111.10 min [22,36,39–41]. Weekday to weekend shifts in sleep are concerning since studies show that the quality of sleep is severely interrupted. For instance, 67.40% of youth report getting less than seven hours of sleep and 41.10% report that they do not get enough sleep [41]. One study reports that 63.20% of youth do not meet the recommended amount of weekday sleep and 50.60% do not meet the recommended amount of weekend sleep [38]. Sleep satisfaction is also inadequate according to self-report, where 74% of youth report that their sleep satisfaction is moderate or unsatisfactory [41] and other studies report that two-thirds of youth get < 8 h sleep per night [35]. A longitudinal study of 343 youth in 10th–12th grade reports a linear decline in weekday sleep of 0.35 h and weekend sleep of 0.45 h from the 10th grade to three years post high school [40]. This suggests that not only are there weekday to weekend shifts in sleep, but children are getting less sleep overall as they age. Perhaps weekend sleep shifts are attributed to later bedtimes. A study reports that school-aged children with a late bedtime have longer sleep onset latency and also sleep one hour less at night compared to children with an early bedtime [34]. In addition, studies show that a greater shift towards more total sleep time on the weekend is associated with greater eating in absence of hunger, which contributes to weight gain and obesity [39]. The literature clearly supports the need for anchoring a regular morning waketime, as this has been shown to be the best way to manage behavioral sleep problems [63]. Future work should also assess nightly screen time [64–68] and sleep hygiene practices [69–72] when assessing sleep duration.

*Strengths and Limitations*

The strength of this study is the objective measurement of sleep and PA collected from an otherwise unknown, isolated, and underserved population. The research team successfully recruited 38 students to participate in this ongoing work to develop the dogjoring program. Conducting a study with this isolated Nunamiut community is a strength in and of itself. Not only does this study show trends that suggest the Nunamiut have adopted a western lifestyle, but this is perhaps more deleterious since there are fewer community resources (e.g., health care systems and health care providers) to address these issues. Although the research team was able to recruit 38 participants, this small sample size limits sensitivity in data analysis. In addition, the cross-sectional analysis provides only descriptive results. Finally, the analysis included overall trends in the data, and the contrast between high levels of PA and a high obesity prevalence suggests that interventions aimed at reducing obesity should address other obesogenic factors in addition to promoting PA. Future work needs to be undertaken with a larger sample size and assessing gender differences in sleep and PA. Sleep assessments should include nightly screen time and sleep hygiene. In spite of these limitations, future work will build upon this work to gain a deeper understanding of sleep and PA in this population.

## 5. Conclusions

This study presents data on sleep, PA, and demographic information from 38 Nunamiut youth and children living in Anaktuvuk Pass in Alaska. Results show that high school youth engage in more sedentary activity, and less light, moderate, and vigorous PA, compared to middle school and elementary school children. This suggests that as children age into youth they move less. All participants demonstrated shifts in favor of less PA, more sedentary activity, and variable sleep patterns on the weekend compared to the weekday. Finally, approximately 11 (40.70%) of the participants were either overweight or obese. Future work should consider the time of year the research is carried out. This is important, especially in the fall, when Nunamiut high school youth are engaged in hunting, and may provide a more realistic look at PA trends in this population. In addition, differences in gender should be examined in order to determine if males and females have different physical activity and sleep needs.

**Author Contributions:** Conceptualization, V.G., J.P. and D.M.; methodology, J.P.; software, V.G. and J.P.; validation, V.G., J.P. and D.M.; formal analysis, J.P.; investigation, J.P. and D.M.; resources, V.G., J.P. and D.M.; data curation, J.P.; writing—original draft preparation, V.G.; writing—review and editing, V.G., J.P. and D.M.; visualization, V.G., J.P. and D.M.; supervision, V.G., J.P. and D.M.; project administration, J.P.; funding acquisition, J.P. and V.G. All authors have read and agreed to the published version of the manuscript.

**Funding:** This research was funded by the National Institute of General Medical Sciences of the National Institutes of Health under Award Numbers UL1GM118991, TL4GM118992, RL5GM118990, P30GM103325, U54GM115371 and U54GM104944. The content is solely the responsibility of the authors and does not necessarily represent the official views of the National Institutes of Health. The University of Alaska is an AA/EO employer and educational institution and prohibits illegal discrimination against any individual.

**Institutional Review Board Statement:** The study was conducted in accordance with the Declaration of Helsinki, and approved by the Institutional Review Board of the University of Alaska Fairbanks (protocol code 1089068 approved on 21 June 2017).

**Informed Consent Statement:** Informed consent was obtained from all subjects involved in the study.

**Data Availability Statement:** The data presented in this manuscript were obtained through a NIH-funded research collaboration with the Nunamiut people of Anaktuvuk Pass in the Brooks Range mountains of Alaska. They were generated through the actions of the research team and in partnership with the people of Anaktuvuk Pass. The community has approved the publication and dissemination of the data in the format provided in this manuscript. The primary dataset from our study is owned by the people of Anaktuvuk Pass, in recognition of tribal sovereign authority stipulated by the government of the United States. Requests for access to the primary data can be made to the Naqsragmiut tribal leadership. While we cannot speak for them, we believe that all requests would be given due consideration. Such requests can be directed to Jacques Philip.

**Acknowledgments:** First and foremost, we would like to thank the community members of Anaktuvuk Pass, the Naqsragmiut Tribal Council and the Nunamiut School for their participation. Additionally, thank you to Brooks Fry and Lars Flora for coaching the youth, Nina Hansen, Erin Trochim and Abigail Fry for their assistance in coaching and data collection and Lisa Wexler for her mentoring.

**Conflicts of Interest:** The authors declare no conflict of interest.

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
