# Peer review of "Physical Activity, Sleep, and Demographic Patterns in Alaska Native Children and Youth Living in Anaktuvuk Pass"

_2673-995X, doi:10.3390/youth3010021_

Round 1
Reviewer 1 Report
Line 53: All the context that the authors have done is very important for the reader, but could some comment be added about sunlight considering its relationship with sleep? Example: What time does it dawn in Anaktuvuk? What time does it get dark? Were children and adolescents able to expose themselves to sunlight?
Line 82-89: very detailed. I suggest that authors reduce the amount of descriptive data and summarize the information.
Line 108-115: This is also very detailed. I suggest just writing the average difference in sleep duration between weekdays and weekends.
Additional comment: I don't know if the data exist, but it would be interesting to make some reference to the time these children and adolescents used the screen due to its direct relationship with sleep.
Lines 157-160: The aim of the study is to describe physical activity, sleep, and weight-related variables in young people. However, the authors inform that the students participated for one hour in a program with physical activities. In addition, they could also participate for two hours after school schedule. Therefore, the present reviewer understands that the authors need to inform if this program was promoted only in the week of data collection, or if it is already a regular practice at the school. Because, if it is unusual, this can characterize a certain interference in the records of minutes in physical activity in the week data collection.
Line 204: Why did the authors not use a sleep recall?
Line 206: It is important that the authors inform how many hours/day the volunteers used the accelerometer after all the data were collected.
Lines 212-215: All text can be written in section “2.6.” or “2.4.”. In section “2.7.” there is still a need for a more detailed analysis. For example, what was the comparison test? Why was the effect size not demonstrated? Did the data follow the requirements for the comparison test? LQMM analysis has important statistical requirements. Did the data suit them? This needs to be better described. In addition, it is interesting to put the version of RStudio that was used.
Line 255: It is unnecessary to say the total number of volunteers, boys AND girls. Just say the total n(%) and that of boys OR girls.
In table 1: two variables are categorized in: not at all, some, and a lot. Such categories were not previously explained in the methods section and it is difficult to understand what each one means. Thus, authors can explain them in section “2.5. Demographics” or in the table caption. However, in the section “2.5. Demographics”would be more appropriate.
In table 2: the authors present a comparison between weekdays and weekends, but the statistical test for the comparison was not mentioned in section “2.7.”
Additional comment: short sleep duration is also related to overweight/obesity. Given the short duration, it would be interesting to test the relationship between sleep and overweight/obesity in young people.
Lines 270-271: I suggest that the authors do not use the term “compared” because there was no comparison between who was overweight, obese, or normal weight. In fact, there was only one description of the prevalences.
Lines 273:297: The paragraph is too long. All information is important and confirms the authors' findings, but it is really important to reorganize the paragraph so that readers can understand it better. In addition, the secular trend is cited twice, once above the paragraph, once below, when, in fact, they could be cited at the same time, further reinforcing what the authors want to inform in a simpler way.Still in this paragraph, it is possible that the importance of this data for the community was missing. Everything is treated in a general way, but what are the harms and benefits of this information for this specific community?
Line 318: remove the comma before “[13]”.
Lines 331-361: Indeed, there is a secular trend towards reduced sleep duration in children of all ages as previously investigated by Lisa Matricciani. But, as previously mentioned, it is important for the authors to indicate whether there are data related to time in front of screens, as this is one of the main factors that interfere with the duration and quality of sleep. In addition, sleep hygiene is a factor that is also related to the same duration and quality. However, it should be noted that ambient luminosity (sunlight, moonlight) and exposure to the sun are also other factors, and when it comes to the location of data collection, it would be important for the authors to point out what this luminosity is like in April.
Additional comment: in fact, the sample size is a limitation, but also a paradox, since few studies show the scientific community the behavior that native communities have concerning physical activity and sleep, in this case.Therefore, and this is, of course, my point of view, that studying this community is also a strength, as one can begin to point out that even more isolated communities are also adopting behaviors of highly industrialized communities and, perhaps, the damage to health is more compromising due to reduced access to health systems, professionals who may be in these communities caring for these people.

Reviewer 2 Report
Dear Authors,
You can find attached the document with the comments derived from the review I have carried out.
Kind regards

Round 2
Reviewer 2 Report
Dear Authors,
Please see attached the review document.
Kind regards
